# Ruling out pulmonary embolism across different healthcare settings: A systematic review and individual patient data meta-analysis

Geert-Jan Geersing[1☯]*, Toshihiko Takada[1,2☯], Frederikus A. Klok[3], Harry R. Büller[4], D. Mark Courtney[5], Yonathan Freund[6], Javier Galipienzo[7], Gregoire Le Gal[8], Waleed Ghanima[9], Jeffrey A. Kline[10], Menno V. Huisman[3], Karel G. M. Moons[1,11], Arnaud Perrier[12], Sameer Parpia[13,14], Helia Robert-Ebadi[12], Marc Righini[12], Pierre-Marie Roy[15], Maarten van Smeden[1], Milou A. M. Stals[3], Philip S. Wells[8], Kerstin de Wit[14,16], Noémie Kraaijpoel[4], Nick van Es[4]

1 Julius Center for Health Sciences and Primary Care, University Medical Center Utrecht, Utrecht University, Utrecht, the Netherlands, 2 Department of General Medicine, Shirakawa Satellite for Teaching And Research (STAR), Fukushima Medical University, Fukushima, Japan, 3 Department of Medicine, Thrombosis and Haemostasis, Dutch Thrombosis Network, Leiden University Medical Center, Leiden, the Netherlands, 4 Department of Medicine, Amsterdam University Medical Center, Amsterdam Cardiovascular Sciences, Amsterdam, the Netherlands, 5 Department of Emergency Medicine, University of Texas Southwestern Medical Center, Dallas, Texas, United States of America, 6 Sorbonne University, Emergency Department, Hôpital Pitié-Salpêtrière, Assistance Publique—Hôpitaux de Paris, Paris, France, 7 Service of Anesthesiology, MD Anderson Cancer Center Madrid, Madrid, Spain, 8 Department of Medicine, University of Ottawa, Ottawa Hospital Research Institute, Ottawa, Canada, 9 Department of Medicine, Østfold Hospital Trust, Norway and Institute of Clinical Medicine, University of Oslo, Oslo, Norway, 10 Department of Emergency Medicine, Wayne State School of Medicine, Detroit, Michigan, United States of America, 11 Cochrane Netherlands, University Medical Center Utrecht, Utrecht University, Utrecht, the Netherlands, 12 Division of Angiology and Hemostasis, Geneva University Hospitals and Faculty of Medicine, Geneva, Switzerland, 13 Department of Oncology, McMaster University, Hamilton, Canada, 14 Department of Health Research Methods, Evidence, and Impact, McMaster University, Hamilton, Canada, 15 UNIV Angers, UMR (CNRS 6015—INSERM 1083) and CHU Angers, Department of Emergency Medicine, F-CRIN InnoVTE, Angers, France, 16 Department of Emergency Medicine, Queen's University, Kingston, Canada

☯ These authors contributed equally to this work.
* G.J.Geersing@umcutrecht.nl

**Data Availability Statement:** Data included in this individual patient data meta-analysis (IPD-MA) cannot be shared publicly without restrictions

## Abstract

### Background

The challenging clinical dilemma of detecting pulmonary embolism (PE) in suspected patients is encountered in a variety of healthcare settings. We hypothesized that the optimal diagnostic approach to detect these patients in terms of safety and efficiency depends on underlying PE prevalence, case mix, and physician experience, overall reflected by the type of setting where patients are initially assessed. The objective of this study was to assess the capability of ruling out PE by available diagnostic strategies across all possible settings.

### Methods and findings

We performed a literature search (MEDLINE) followed by an individual patient data (IPD) meta-analysis (MA; 23 studies), including patients from self-referral emergency care (*n* =

because this study includes human research participant data with different ethical (e.g. differences in provided informed consent) and legal restrictions applying based upon the setting and country where primary data underlying this IPD-MA originated. Thus, prior to sharing data, a discussion within the steering group of collaborative investigators for this IPD-MA, as well as consulting an ethics committee where appropriate, is needed to ensure that data are shared in accordance with participant consent and all applicable local laws. Data are available from the Institutional Data Access (contact via SecretariaatHAG-Onderzoek@umcutrecht.nl) for researchers who meet the criteria for access to confidential data, accompanied with a protocol describing the research questions aimed to answer. URL: https://juliuscentrum.umcutrecht.nl/en/.

**Funding:** GJG is supported by a personal Vidi grant from the Dutch Research Council (grant number 91719304). URL: https://www.nwo.nl/en/calls/nwo-talent-programme. The funders had no role in study design, data collection and analysis, decision to publish, or preparation of the manuscript.

**Competing interests:** I have read the journal's policy and the following authors of this manuscript have the following competing interests: FAK reports research grants from Bayer, Bristol-Myers Squibb, Boehringer-Ingelheim, MSD, Daiichi-Sankyo, Actelion, the Dutch thrombosis association, The Netherlands Organisation for Health Research and Development and the Dutch Heart foundation. GLG holds the Chair on Diagnosis of Venous Thromboembolism from the Department of Medicine, University of Ottawa and a Clinician Scientist award from the Heart and Stroke Foundation of Canada. WG reports advisory board participation from Amgen, Novartis, Pfizer, Principia Biopharma Inc, a Sanofi Company, and Sanofi, SOBI, Griffols, UCB; lecture honoraria from Amgen, Novartis, and Pfizer; and grants from Bayer, Bristol Myers Squibb, SOBI, Griffols and Pfizer. No other disclosures were reported.

**Abbreviations:** CI, confidence interval; CPTA, computed tomography pulmonary angiography; IPD, individual patient data; ISTH, International Society on Thrombosis and Haemostasis; MA, meta-analysis; PE, pulmonary embolism; PERC, Pulmonary Embolism Rule-out Criteria; PI, prediction intervals; PRISMA-DTA, Preferred Reporting Items for Systematic Reviews and Meta-Analyses of Diagnostic Test Accuracy; PRISMA-IPD, Preferred Reporting Items for Systematic Reviews and Meta-Analyses of Individual

12,612), primary healthcare clinics (*n* = 3,174), referred secondary care (*n* = 17,052), and hospitalized or nursing home patients (*n* = 2,410). Multilevel logistic regression was performed to evaluate diagnostic performance of the Wells and revised Geneva rules, both using fixed and adapted D-dimer thresholds to age or pretest probability (PTP), for the YEARS algorithm and for the Pulmonary Embolism Rule-out Criteria (PERC). All strategies were tested separately in each healthcare setting. Following studies done in this field, the primary diagnostic metrices estimated from the models were the "failure rate" of each strategy—i.e., the proportion of missed PE among patients categorized as "PE excluded" and "efficiency"—defined as the proportion of patients categorized as "PE excluded" among all patients. In self-referral emergency care, the PERC algorithm excludes PE in 21% of suspected patients at a failure rate of 1.12% (95% confidence interval [CI] 0.74 to 1.70), whereas this increases to 6.01% (4.09 to 8.75) in referred patients to secondary care at an efficiency of 10%. In patients from primary healthcare and those referred to secondary care, strategies adjusting D-dimer to PTP are the most efficient (range: 43% to 62%) at a failure rate ranging between 0.25% and 3.06%, with higher failure rates observed in patients referred to secondary care. For this latter setting, strategies adjusting D-dimer to age are associated with a lower failure rate ranging between 0.65% and 0.81%, yet are also less efficient (range: 33% and 35%). For all strategies, failure rates are highest in hospitalized or nursing home patients, ranging between 1.68% and 5.13%, at an efficiency ranging between 15% and 30%. The main limitation of the primary analyses was that the diagnostic performance of each strategy was compared in different sets of studies since the availability of items used in each diagnostic strategy differed across included studies; however, sensitivity analyses suggested that the findings were robust.

## Conclusions

The capability of safely and efficiently ruling out PE of available diagnostic strategies differs for different healthcare settings. The findings of this IPD MA help in determining the optimum diagnostic strategies for ruling out PE per healthcare setting, balancing the trade-off between failure rate and efficiency of each strategy.

## Author summary

### Why was this study done?

- Pulmonary embolism (PE; i.e., clots in pulmonary vessels) is a potentially fatal condition, and patients suspected of having this condition are encountered in many different healthcare settings.

- To help physicians with ruling out PE without additional imaging tests, several diagnostic strategies exist, consisting of clinical items and a blood test (D-dimer testing), with different approaches to interpret this D-dimer test, i.e., using a fixed threshold, an age-adjusted manner, or adjusting D-dimer interpretation to a pretest probability (PTP) of PE.

Participant Data; PTP, pretest probability; VTE, venous thromboembolism.

- However, it remains unknown how each diagnostic strategy performs in different healthcare settings like emergency care, primary healthcare, secondary hospital care, and inpatient care.

### What did the researchers do and find?

- The researchers searched and collected individual patient data (IPD) of existing studies that can be used to evaluate the performance of diagnostic strategies to exclude the possibility of PE.

- By analyzing the data of over 35,000 patients suspected of PE from 23 studies, the researchers validated the performance of diagnostic strategies for suspected PE across different healthcare settings.

- In healthcare settings with a higher prevalence of PE—compared to those with a lower prevalence—each diagnostic strategy tended to miss more patients with PE (i.e., less safe) and identified less patients in whom PE could be ruled out without imaging (i.e., less efficient), notably for strategies with a variable D-dimer interpretation.

### What do these findings mean?

- The performance of diagnostic strategies varied considerably across different healthcare settings due to the difference in patient characteristics and prevalence of PE.

- Our findings can be used to choose the optimum diagnostic strategies in each healthcare setting, balancing the trade-off between decreasing unnecessary imaging studies and missing patients with PE.

## Introduction

Pulmonary embolism (PE) is one of the most difficult diagnoses in clinical medicine, encountered daily in a variety of healthcare settings [1,2]. Due to potentially fatal consequences of missing PE [3,4], physicians tend to perform diagnostic imaging tests even when PE is considered not the most likely diagnosis. Some argue against this low threshold for diagnostic workup since such overtesting can lead to unnecessary radiation exposure, cost, and potential adverse events related to the use of contrast media [5]. At the same time, it has been argued that PE should be suspected more often to prevent potentially life-threatening delay in diagnosis [6].

To help physicians with this clinical dilemma, various diagnostic strategies for ruling out PE have been developed over time, all consisting of a set of clinical variables that are often combined with a blood test to detect clot degradation, i.e., D-dimer [7,8]. Given the differences in case mix and underlying prevalence of PE, it is likely that each diagnostic strategy has different merits across different healthcare settings [9,10]. Nevertheless, evidence on the performance of the currently available diagnostic strategies across different healthcare settings is limited, notably for settings like primary healthcare or inpatient care.

Hence, we performed a comprehensive systematic review followed by an individual patient data (IPD) meta-analysis (MA) to explore the performance of diagnostic strategies for PE across a variety of healthcare settings. The secondary aim of this study was to investigate the relationship between PE prevalence and the diagnostic performance measures of each strategy.

## Methods

Throughout this paper, we adhere to the Preferred Reporting Items for Systematic Reviews and Meta-Analyses of Individual Participant Data (PRISMA-IPD) and Preferred Reporting Items for Systematic Reviews and Meta-Analyses of Diagnostic Test Accuracy (PRISMA-DTA) guidance on systematic reviews including IPD, where applicable [11,12]. The checklists are available in Tables A, B, and C in S1 Checklists. Ethical approval including written informed consent was obtained in each original study, and analyses described in this paper on optimizing diagnostic strategies for suspected PE were aligned with the informed consent as provided by individual patients in each study. Therefore, no additional ethical approval was required for this MA.

### Protocol registration

This study was preregistered in the PROSPERO registration (see https://www.crd.york.ac.uk/prospero ID 89366), and the protocol has been published [13].

### Diagnostic strategies under evaluation

Based on a previous systematic review [14] and discussion among experts, we a priori selected 11 existing diagnostic strategies under evaluation. The overview of these index strategies is shown in Table A in S1 Text. The 2 most commonly used clinical decision rules for pretest probability (PTP) assessment, the Wells and revised Geneva rules [14], are to be combined with D-dimer testing, with D-dimer interpretations either using a fixed cutoff (using either qualitative or quantitative D-dimer testing), adjusted to PTP, or adjusted to age [15,16]. The YEARS algorithm is a simplified version of the Wells rule with PTP-adjusted D-dimer [17]. The Pulmonary Embolism Rule-out Criteria (PERC) algorithm, which comprises 8 clinical items, was also evaluated [18]. This strategy differs from the other diagnostic strategies as it was originally developed for excluding PE in patients with a low clinical impression of PE. Hereto, following earlier studies, the PERC algorithm was validated in combination with (i) a Wells rule of 4 points or less; or (ii) physician's gestalt considering PE unlikely ("low gestalt"). The PERC algorithm could only be evaluated for the settings "self-referral" emergency care and referred secondary care due to missing information on oxygen saturation in most of the studies in the other settings.

### Study eligibility, identification, and selection

The process of study selection for the IPD-MA was described in detail in the protocol [13]. In short, to retrieve eligible studies, MEDLINE was first searched from January 1, 1995 to August 25, 2016 (this was recently updated until November 1, 2021). Studies were eligible if they (1) had a prospective or cross-sectional design and included patients with clinically suspected PE (in diagnostic research of venous thromboembolism [VTE], prospective cohort studies are common because VTE is often defined by clinical follow-up in patients whom a PTP of VTE is deemed unlikely); (2) assessed the variables to validate at least one of the diagnostic strategies under evaluation; (3) included a clear description of the source of patient enrolment or clinical healthcare setting; (4) objectively confirmed VTE diagnosis (i.e., PE or deep vein thrombosis)

with an established reference test method (either imaging [computed tomography pulmonary angiography (CTPA), ventilation–perfusion lung scan, or digital subtraction angiography] or clinical follow-up of at least 1 month); and (5) included at least 50 patients with confirmed VTE. Full-text screening was performed independently by 2 couples of authors (GJG and NK and FAK and NvE), and 40 potentially eligible papers were identified. With all principal investigators from these 40 retrieved studies invited, the results of this literature search were discussed during a meeting at the International Society on Thrombosis and Haemostasis (ISTH) conference in Berlin in 2017. The search results were complimented by asking those experts in the field of diagnosing VTE about whether they knew any additional datasets eligible for this IPD.

## Risk of bias assessment across studies

Three pairs of authors (GJG and TT, NvE and NK, and FAK and MAMS), who were not involved in the original studies, independently assessed each eligible study for potential sources of bias and applicability concerns using the QUADAS-2 tool [19]. Any disagreements were solved by discussion within each pair and subsequently between the pairs.

## Healthcare settings

We defined the following 4 categories of healthcare settings in which patients suspected of PE are typically encountered:

i. Self-referral emergency care: Patients typically present themselves without a referral by a general physician or specialist. This setting is characterized by a (very) low PE prevalence (i.e., around 5%) among patients with clinically suspected PE and has relatively good access to additional imaging or laboratory workup. Given that the studies performed in this setting emphasized on preselection of patients who need to undergo D-dimer testing, thus not explicitly to evaluate a clinical decision rule for patients with a clear suspicion of PE, we only validated the PERC algorithm in this setting.

ii. Primary healthcare: Outpatient or community healthcare clinics where patients are investigated by a general practitioner, family doctor, or general internist who needs to decide on the need for further referral or diagnostic testing, with relatively restricted access to laboratory or imaging workup. The PE prevalence is usually low to intermediate (i.e., between 5% and 15%).

iii. Referred secondary care: In this setting, patients are referred (mostly by general practitioners, family doctors, or general internists) based upon a clear clinical suspicion of PE. In this setting, the PE prevalence in suspected patients is intermediate to relatively high (i.e., between 15% and 25%).

iv. Hospitalized or nursing home care: In this setting, patients are either hospitalized or in nursing homes, reflecting more severe and progressive illness with a high risk of PE. PE prevalence in the suspected population is typically high (i.e., above 25%).

To categorize each study into 1 of the 4 settings, expert panel members (GJG, FAK, MAMS, NK, and NvE) independently grouped each study and discussed disagreements until they reached a consensus. For studies that were performed in more than 1 setting (e.g., including both outpatients and inpatients), each patient was categorized based on the information provided by the principal investigators.

## Data collection and harmonization

Principal investigators of eligible studies were asked to provide their original, anonymized datasets. These datasets were then harmonized by adjusting coding and definition of each variable using a template developed for this IPD-MA; see Table B in S1 Text.

## Outcomes

The primary outcomes were diagnostic indices, i.e., failure rate and efficiency of each diagnostic strategy across different healthcare settings. Failure rate, which is a frequently applied measure for diagnostic safety in the VTE domain, was defined as the proportion of missed PE patients among those categorized as "PE excluded" by each diagnostic strategy. Efficiency of a strategy was defined as the proportion of patients categorized by the strategy as "PE excluded" among all patients. Additionally, we also estimated the traditional diagnostic indices, sensitivity and specificity.

## Missing data

Summary of missing data in each study is shown in Table C in S1 Text. Within each study, missing values were imputed using multiple imputation techniques with chained equations with all available variables, except for variables missing in more than 80% of patients in the study [20]. The detail of imputation procedure is described in S1 Text.

## Statistical analyses

The statistical analysis plan is described in detail in S1 Text. To evaluate the diagnostic performance of each strategy across different healthcare settings, we used multilevel logistic regression models [21,22]. In models for failure rate and efficiency, a random effect for the intercept was applied to account for clustering of observations within studies. In models for sensitivity and specificity, we used univariate random effects modeling due to nonconvergence issues encountered in bivariate random effects modeling [23]. By using these models, the diagnostic performance measures were estimated with 95% confidence intervals (CIs). In addition, between-study heterogeneity was assessed by calculating 95% prediction intervals (PIs), which indicates the performance that can be expected when the diagnostic strategy is applied in a new study [24]. Forest plots were drawn to visualize the failure rate and efficiency for the different strategies across different healthcare settings. In addition, the range of failure rate and efficiency of each diagnostic strategy in included studies was visualized with $I^2$ [25].

Although our primary aim was to evaluate the performance of diagnostic strategies across different healthcare settings, the categorization of healthcare settings by the expert panel might still be arbitrary. Therefore, we assessed the relationship between failure rate and efficiency with underlying PE prevalence in each study as well, as this was deemed one of the most important distinctive characteristics of different healthcare settings. In accordance with a previous systematic review [26], log-transformed prevalence was added as a continuous covariable to the aforementioned multilevel logistic regression models. The relationship between PE prevalence and failure rate or efficiency of each strategy was plotted to graphically illustrate the impact of PE prevalence on these outcomes.

Finally, given that the availability of items used in each diagnostic strategy differed across included studies, the diagnostic performance of each strategy was estimated in different sets of studies. This inherently makes comparisons of each strategy indirect, and, therefore, we performed additional sensitivity analyses including only studies in which all diagnostic strategies can be calculated. Such an analysis yields a direct comparison among diagnostic strategies.

All analyses were performed using R, version 3.6.3 (R foundation for Statistical Computing, www.R-project.org), particularly using the lme 4 package.

## Results

The systematic literature search identified 3,892 unique studies [13]. After applying the eligibility criteria and scrutinizing original data files and publications, a total of 23 studies were selected to be included in this IPD-MA for a total of 35,248 unique patients suspected of PE; see Fig A in S1 Figs. Risk of bias of included studies was generally scored as low; see Fig B in S1 Figs.

### Study and patient characteristics

A summary of the included studies is shown in Table D in S1 Text. Studies were published between 2000 and 2019. A total of 5 studies were conducted in self-referral emergency care (*N* = 12,612; mean prevalence 7%), 4 in primary healthcare (*N* = 3,174; mean prevalence 9%), 14 in referred secondary care (*N* = 17,052; mean prevalence 20%), and 9 studies included patients hospitalized or in nursing home (*N* = 2,410; mean prevalence 24%). Detailed patient characteristics in each healthcare setting are shown in Table 1.

### Accuracy of different diagnostic strategies across healthcare settings

Fig 1 shows the failure rate and efficiency of the diagnostic strategies across healthcare settings. The range of failure rate and efficiency in the included studies are shown with $I^2$ in Fig C in S1 Figs. Sensitivity and specificity of the 11 diagnostic strategies across healthcare settings are shown in Table 2. All strategies had a sensitivity higher than 90% in all settings (range: 93.3% to 99.6%), while specificity decreased in healthcare settings with higher PE prevalence (range: 7.9% to 67.4%).

### Self-referral emergency care

The PERC algorithm was evaluated in combination with a Wells rule ≤4 points or "low gestalt." Failure rate was 1.12% (95% CI 0.74 to 1.70) for the PERC algorithm combined with a Wells rule ≤4 points and 0.90% (95% CI 0.54 to 1.48) for that with "low gestalt." Efficiency was higher for the PERC algorithm combined with a Wells rule ≤4 points (21%) than when that with "low gestalt" (13%).

### Primary healthcare

The failure rate ranged from 0.13% (95% CI 0.03 to 0.62) for the Wells rule with a fixed D-dimer cutoff to 0.69% (95% CI 0.31 to 1.52) for the Wells rule with a qualitative or fixed D-dimer cutoff, while efficiency ranged from 38% (95% CI 25 to 52) for the Wells rule with a fixed D-dimer cutoff to 62% (95% CI 48 to 74) for the Wells rule with PTP-adjusted D-dimer.

### Referred secondary care

In general, strategies with PTP-adjusted D-dimer (i.e., YEARS and Wells or revised Geneva rule combined with PTP-adjusted D-dimer) showed a higher failure rate than the others without overlapping in their 95% CIs: Failure rate was 2.10% (95% CI 1.59 to 2.75) for YEARS, 3.06% (95% CI 2.47 to 3.78) for the Wells rule with PTP-adjusted D-dimer, and 2.95% (95% 2.34 to 3.71) for the revised Geneva rule with PTP-adjusted D-dimer, respectively. Among the others, the failure rate ranged from 0.32% (95% CI 0.17 to 0.60) to 1.17% (95% CI 0.79 to

**Table 1. Patient characteristics across different healthcare settings.**

| | Missing proportion[a] | Self-referral emergency care | | | Primary healthcare | | | Referred secondary care | | | Hospitalized or nursing home care | | |
|---|---|---|---|---|---|---|---|---|---|---|---|---|---|
| | | Patients without PE N = 11,682 | Patients with PE N = 930 | Total N = 12,612 | Patients without PE N = 2,890 | Patients with PE N = 284 | Total N = 3,174 | Patients without PE N = 13,610 | Patients with PE N = 3,442 | Total N = 17,052 | Patients without PE N = 1,831 | Patients with PE N = 579 | Total N = 2,410 |
| Age (years) | 0.0 | 46.0 (35.0, 59.0) | 55.0 (41.0, 69.0) | 47.0 (36.0, 60.0) | 50.4 (36.0, 63.2) | 56.1 (44.0, 70.7) | 51.0 (36.8, 64.0) | 56.0 (41.2, 70.0) | 64.1 (50.0, 76.0) | 57.4 (43.0, 71.7) | 58.5 (44.6, 71.0) | 63.6 (50.9, 74.2) | 60.0 (46.0, 72.1) |
| Female sex | 0.0 | 8,163 (69.9) | 541 (58.1) | 8,704 (69.0) | 1,973 (68.3) | 170 (59.9) | 2,143 (67.5) | 8,143 (59.8) | 1,781 (51.7) | 9,924 (58.2) | 1,121 (61.2) | 335 (57.9) | 1,456 (60.4) |
| Previous VTE | 0.0 | 1,127 (9.6) | 246 (26.5) | 1,373 (10.9) | 249 (8.6) | 63 (22.2) | 312 (9.8) | 1,657 (12.2) | 933 (27.1) | 2,590 (15.2) | 180 (9.8) | 113 (19.5) | 293 (12.2) |
| Heart rate >100 | 0.0 | 3,465 (29.7) | 393 (42.3) | 3,858 (30.6) | 787 (27.2) | 112 (39.4) | 899 (28.3) | 6,203 (24.5) | 1,380 (31.6) | 7,583 (25.6) | 554 (30.2) | 204 (35.3) | 758 (31.5) |
| Surgery or immobilization <4 weeks | 0.0 | 1,932 (16.5) | 252 (27.1) | 2,184 (17.3) | 264 (9.1) | 62 (21.8) | 326 (10.3) | 1,625 (11.9) | 774 (22.5) | 2,399 (14.1) | 640 (34.9) | 301 (52.0) | 941 (39.0) |
| Hemoptysis | 0.0 | 323 (2.8) | 44 (4.7) | 367 (2.9) | 116 (4.0) | 22 (7.7) | 138 (4.3) | 599 (4.4) | 228 (6.6) | 827 (4.8) | 73 (4.0) | 28 (4.9) | 101 (4.2) |
| Active cancer | 0.0 | 860 (7.4) | 153 (16.4) | 1,013 (8.0) | 219 (7.6) | 45 (15.8) | 264 (8.3) | 1,261 (9.3) | 488 (14.2) | 1,749 (10.3) | 297 (16.2) | 139 (24.1) | 436 (18.1) |
| Clinical signs of DVT | 0.0 | 820 (7.0) | 215 (23.1) | 1,035 (8.2) | 200 (6.9) | 79 (27.8) | 279 (8.8) | 668 (4.9) | 668 (19.4) | 1,336 (7.8) | 94 (5.1) | 90 (15.6) | 184 (7.6) |
| Alternative diagnosis less likely than PE | 7.0 | 2,339 (21.4) | 356 (47.8) | 2,695 (23.1) | 826 (28.6) | 180 (63.4) | 1,006 (31.7) | 5,787 (46.4) | 1,902 (62.4) | 7,689 (49.5) | 822 (44.9) | 438 (75.7) | 1,260 (52.3) |
| Quantitative D-dimer (ng/ml) | 15.0 | 328.0 (214.0, 710.0) | 2,234.0 (757.0, 4,000.0) | 350.0 (220.0, 826.0) | 440.0 (270.0, 940.0) | 3,260.0 (1,647.5, 4,000.0) | 490.0 (270.0, 1,160.0) | 606.0 (300.0, 1,128.0) | 2,750.0 (1,300.0, 5,000.0) | 800.0 (363.0, 1,738.9) | 1,000.0 (499.0, 2,300.0) | 3,195.0 (1,573.0, 5,800.0) | 1,352.0 (600.0, 3,110.0) |

Values are median (interquartile range) for continuous variables and numbers (percentages) for categorical variables.

[a]Missing proportion after imputation within each study.

DVT, deep vein thrombosis; N, number of patients; PE, pulmonary embolism; VTE, venous thromboembolism.

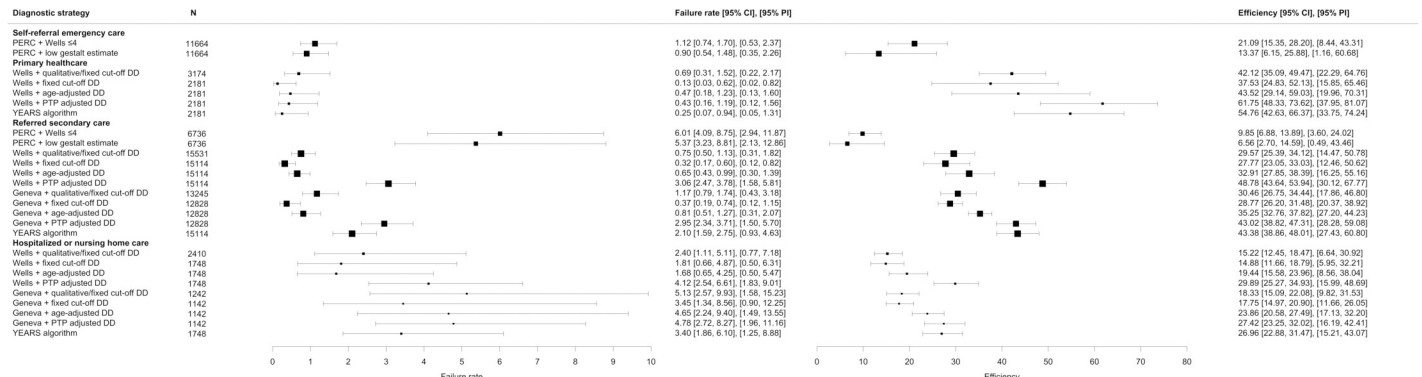

**Fig 1. Forest plot of failure rate and efficiency of the diagnostic strategies across healthcare settings.** CI, confidence interval; (C)PTP, (clinical) pretest probability; DD, D-dimer; *N*, number of patients; PERC, Pulmonary Embolism Rule-out Criteria; PI, prediction interval; PTP, pretest probability.

1.74). Efficiency of the strategies using PTP-adjusted D-dimer was higher than the others without overlapping in their 95% CIs.

Evaluation of the PERC algorithm in combination with a Wells rule of ≤4 points yielded a failure rate of 6.01% (95% CI 4.09 to 8.75) with a corresponding efficiency of 10% (95% CI 7 to 14).

## Hospitalized or nursing home care

The failure rate ranged from 1.68% (95% CI 0.65 to 4.25) for the Wells rule with age-adjusted D-dimer to 5.13% (95% CI 2.57 to 9.93) for the revised Geneva rule with a qualitative or fixed D-dimer cutoff, while efficiency ranged from 15% (95% CI 12 to 19) for the Wells rule with a fixed D-dimer cutoff to 30% (95% CI 25 to 35) for the Wells rule with PTP-adjusted D-dimer. The failure rate of all strategies showed wide overlapping 95% CIs.

## Association between PE prevalence and failure rate/efficiency of diagnostic strategies under evaluation

The relationship between PE prevalence and failure rate or efficiency is visualized in Figs 2 and 3, respectively. In general, as PE prevalence increased, both failure rate and efficiency became poorer (i.e., higher failure rate and lower efficiency).

## Sensitivity analyses allowing direct comparisons

Two sensitivity analyses were performed for direct comparisons. First, we included only patients in whom all diagnostic strategies can be calculated. Due to the lack of studies allowing for such a direct comparison of all strategies, we could include only referred secondary care patients in this sensitivity analysis (*N* = 6,736). Second, as the PERC algorithm is different from the other strategies as it is used in only patients with a very low PTP, we have also included patients in whom all diagnostic strategies except the PERC algorithm can be calculated (including *N* = 11,307 in the referred secondary care and *N* = 1,142 in hospitalized or nursing home care). In both types of sensitivity analyses, we found very similar inferences which supported the robustness of the primary analyses; see Figs D and E in S1 Figs.

## Discussion

In this large, comprehensive international study including over 35,000 patients suspected of PE in various healthcare settings, we validated the performance of diagnostic strategies for

**Table 2. Sensitivity and specificity of diagnostic strategies across healthcare settings.**

| Diagnostic strategy | N | Sensitivity [95% CI], [95% PI] | Specificity [95% CI], [95% PI] |
|---|---|---|---|
| **Self-referral emergency care** | | | |
| PERC + Wells ≤4 | 11,664 | 95.69 [93.93, 96.95], [93.40, 97.20] | 22.23 [16.36, 29.41], [9.22, 44.27] |
| PERC + low gestalt estimate | 11,664 | 96.94 [95.41, 97.97], [94.93, 98.17] | 14.30 [6.34, 28.19], [1.15, 64.07] |
| **Primary healthcare** | | | |
| Wells + qualitative/fixed cutoff DD | 3,174 | 96.39 [85.97, 99.29], [56.48, 99.92] | 49.40 [42.32, 56.50], [29.60, 69.39] |
| Wells + fixed cutoff DD | 2,181 | 99.26 [93.93, 99.91], [91.11, 99.94] | 40.66 [27.61, 55.13], [18.96, 66.61] |
| Wells + age-adjusted DD | 2,181 | 96.84 [89.67, 99.10], [83.64, 99.48] | 47.40 [32.29, 62.99], [24.01, 71.96] |
| Wells + PTP-adjusted DD | 2,181 | 97.11 [92.16, 98.97], [90.81, 99.14] | 67.40 [55.01, 77.79], [46.12, 83.38] |
| YEARS algorithm | 2,181 | 98.20 [92.11, 99.61], [89.47, 99.72] | 60.55 [48.43, 71.52], [39.90, 78.06] |
| **Referred secondary care** | | | |
| PERC + Wells ≤4 | 6,736 | 97.56 [96.61, 98.25], [96.33, 98.39] | 12.00 [8.52, 16.62], [4.59, 27.65] |
| PERC + low gestalt estimate | 6,736 | 98.63 [97.86, 99.12], [97.62, 99.21] | 7.85 [3.15, 17.55], [0.54, 49.44] |
| Wells + qualitative/fixed cutoff DD | 15,531 | 98.38 [95.87, 99.41], [75.51, 99.95] | 36.89 [32.53, 41.47], [20.57, 56.78] |
| Wells + fixed cutoff DD | 15,114 | 99.59 [99.10, 99.82], [98.54, 99.89] | 35.21 [30.19, 40.57], [18.21, 56.84] |
| Wells + age-adjusted DD | 15,114 | 98.93 [98.15, 99.39], [96.21, 99.71] | 41.58 [36.42, 46.93], [24.05, 61.47] |
| Wells + PTP-adjusted DD | 15,114 | 93.25 [91.91, 94.38], [90.02, 95.48] | 60.80 [56.24, 65.19], [43.69, 75.66] |
| Geneva + qualitative/fixed cutoff DD | 13,245 | 97.75 [93.86, 99.27], [64.77, 99.96] | 39.25 [34.57, 44.14], [22.96, 58.28] |
| Geneva + fixed cutoff DD | 12,828 | 99.53 [98.88, 99.80], [97.39, 99.92] | 37.23 [34.00, 40.57], [26.44, 49.45] |
| Geneva + age-adjusted DD | 12,828 | 98.51 [97.37, 99.16], [93.48, 99.68] | 45.27 [42.63, 47.95], [36.72, 54.11] |
| Geneva + PTP-adjusted DD | 12,828 | 94.18 [92.70, 95.38], [89.64, 96.81] | 54.49 [50.82, 58.12], [41.42, 66.98] |
| YEARS algorithm | 15,114 | 96.15 [94.87, 97.12], [91.82, 98.24] | 54.39 [49.87, 58.85], [37.97, 69.93] |
| **Hospitalized or nursing home care** | | | |
| Wells + qualitative/fixed cutoff DD | 2,410 | 99.04 [96.61, 99.75], [80.90, 99.98] | 20.06 [16.79, 23.78], [9.87, 36.34] |
| Wells + fixed cutoff DD | 1,748 | 99.18 [95.95, 99.84], [94.04, 99.89] | 19.82 [15.94, 24.36], [9.02, 37.87] |
| Wells + age-adjusted DD | 1,748 | 99.07 [97.06, 99.71], [94.98, 99.83] | 26.06 [21.49, 31.19], [13.34, 44.47] |
| Wells + PTP-adjusted DD | 1,748 | 95.64 [92.85, 97.38], [91.68, 97.77] | 39.50 [34.27, 44.98], [24.33, 56.96] |
| Geneva + qualitative/fixed cutoff DD | 1,242 | 98.54 [95.00, 99.63], [70.64, 99.98] | 25.82 [21.26, 30.97], [13.55, 43.45] |
| Geneva + fixed cutoff DD | 1,142 | 98.58 [93.10, 99.73], [87.20, 99.86] | 24.47 [20.65, 28.74], [15.92, 35.64] |

(*Continued*)

**Table 2.** (Continued)

| Diagnostic strategy | N | Sensitivity [95% CI], [95% PI] | Specificity [95% CI], [95% PI] |
|---|---|---|---|
| Geneva + age-adjusted DD | 1,142 | 97.18 [92.40, 99.00], [85.07, 99.54] | 32.48 [28.25, 37.02], [24.32, 41.86] |
| Geneva + PTP-adjusted DD | 1,142 | 95.73 [92.06, 97.75], [89.78, 98.29] | 37.29 [32.48, 42.36], [25.44, 50.87] |
| YEARS algorithm | 1,748 | 96.94 [94.31, 98.37], [91.93, 98.88] | 35.83 [30.90, 41.08], [21.98, 52.48] |

CI, confidence interval; DD, D-dimer; N, number of patients; PERC, Pulmonary Embolism Rule-out Criteria; PI, prediction interval; PTP, pretest probability.

suspected PE. We observed that the performance of these strategies varied considerably across different healthcare settings, likely due to the difference in case mix and (thus) PE prevalence. Our findings provide strong evidence on the optimum diagnostic strategies for PE suspicion per care setting, balancing the trade-off between missing PE cases and decreasing unnecessary referrals or follow-up.

## Clinical implications

Our interpretation of the findings is as follows. The PERC algorithm is safe in self-referral emergency care, allowing to preclude additional testing for PE (notably including D-dimer) in about 1 in every 5 patients when combined with a low clinical impression of PE being present, which confirms previous findings [27,28]. In the other settings, as this algorithm appears not to be safe, the use of a diagnostic strategy followed by D-dimer testing is preferred.

In primary healthcare, strategies with PTP-adjusted D-dimer showed equal safety and higher efficiency than those with a fixed or age-adjusted D-dimer cutoff, making them overall an attractive diagnostic strategy. However, in referred secondary care, strategies with PTP-

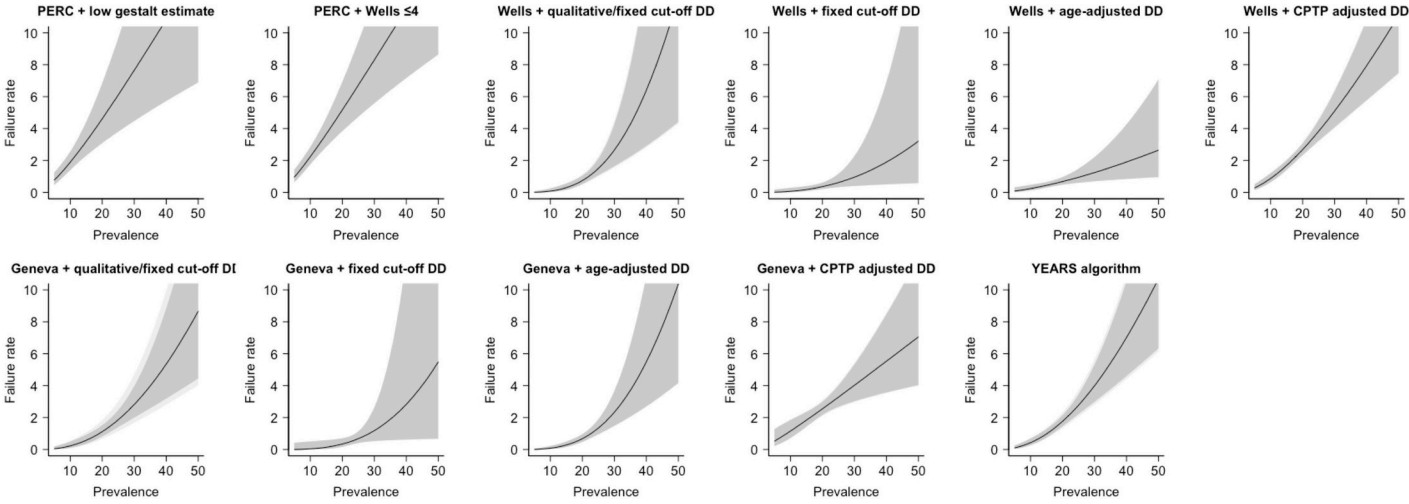

**Fig 2. The relationship between the prevalence of PE and failure rate of each diagnostic strategy.** Gray shaded area shows 95% CI, and light gray shaded area shows 95% PI. CI, confidence interval; (C)PTP, (clinical) pretest probability; DD, D-dimer; PE, pulmonary embolism; PERC, Pulmonary Embolism Rule-out Criteria; PI, prediction interval; PTP, pretest probability.

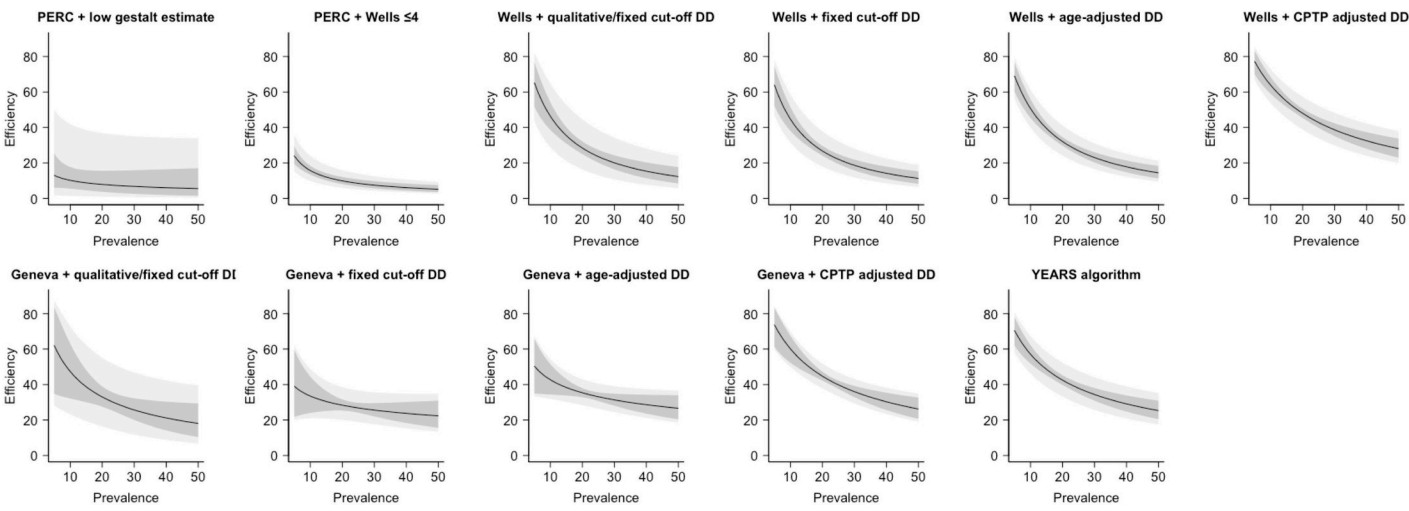

**Fig 3. The relationship between the prevalence of PE and efficiency of each diagnostic strategy.** Gray shaded area shows 95% CI, and light gray shaded area shows 95% PI. CI, confidence interval; (C)PTP, (clinical) pretest probability; DD, D-dimer; PE, pulmonary embolism; PERC, Pulmonary Embolism Rule-out Criteria; PI, prediction interval; PTP, pretest probability.

adjusted D-dimer also had a better efficiency but showed a considerably higher failure rate—ranging between 2.10% and 3.06%—compared to those with age-adjusted D-dimer, which ranged from 0.65% to 0.81%.

Finally, in hospitalized or nursing home care, the observed failure rate was higher than that for the other settings, ranging between 1.81% and 5.13%. Moreover, as clearly observed in wide 95% CIs and PIs, the precision of our inferences was not sufficient to draw firm conclusions in this setting.

When deciding what diagnostic strategy to use, it should be acknowledged that no diagnostic strategy in patients suspected of PE will be completely safe, i.e., yielding a "failure rate" of 0%. In fact, even CTPA, which is used as the "reference standard" for PE in modern clinical medicine, is not perfectly safe as the cumulative VTE incidence at 3 months after a normal CTPA—i.e., the "failure rate" of CTPA—was reported to be 1.20% (95% CI 0.48 to 2.60) [29]. Accordingly, it could be argued that any diagnostic strategy with a failure rate around 1% to 2% is as safe as referring all patients for CTPA, and this safety threshold is generally considered the adequate standard provided by the ISTH. Nevertheless, this safety threshold is dependent on case mix, exemplified by a higher cumulative VTE incidence at 3 months following a normal CTPA in patients with a high PTP (6.3%; i.e., patients with risk factors such as cancer, previous VTE, and immobilization). Thus, the acceptable threshold of a failure rate could be higher in healthcare settings that include more high-risk patients (i.e., high PE prevalence) than in those including more low-risk patients (i.e., low PE prevalence). Such a prevalence-adjusted threshold of failure rate indeed has been proposed by the ISTH [9]. If this was applied to each healthcare setting in this IPD-MA for illustrative purposes, the acceptable threshold of failure rate should range between 0.71% and 1.86% in self-referral emergency care, between 0.72% and 1.87% in primary healthcare, between 0.78% and 1.93% in referred secondary care, and between 0.80% and 1.95% in hospitalized or nursing home care, respectively. In that case, the optimum strategy (i.e., most efficient strategy with acceptable failure rate) may be the PERC algorithm in emergency care, a PTP-adjusted D-dimer strategy in primary healthcare, and an age-adjusted strategy in referred secondary care, while no strategy showed an acceptable failure rate in hospitalized or nursing home care.

Nevertheless, as these prevalence-adjusted thresholds are proposed only for planning diagnostic studies rather than for the use in clinical practice [9], physicians need to set the acceptable threshold of failure rate for their own setting and standards and subsequently choose the optimum diagnostic strategy, likely dictated by clinical context. We believe that our findings can be used to aid that clinical decision-making, balancing the trade-off between safety and efficiency, and tailored to the specific setting and case mix where they work and encounter patients suspected of PE. Furthermore, by combining with various factors (e.g., patient perceptions and demands, availability of imaging studies, and benefit/cost associated with different recommendations) in a clinical setting where it is applied, our findings could be a useful basis for developing a clinical guideline for the diagnosis of PE.

This large-scale international study included over 35,000 patients suspected of PE, coming from a variety of healthcare settings. In addition, we used state-of-the-art statistical methods to quantify diagnostic performance of currently available diagnostic strategies. For full appreciation, some aspects of this study though need specific attention.

First, the availability of items used in each diagnostic strategy differed across included studies. As such, in the primary analyses, the diagnostic performance of each strategy was compared in different sets of studies. Accordingly, we added the sensitivity analyses for a direct comparison of the diagnostic strategies, which yielded very similar results supporting the robustness of the primary analyses.

Second, although we defined the categorization of healthcare settings through profound discussion among expert panel members, it could still be arbitrary. Thus, we analyzed the relationship between failure rate or efficiency and PE prevalence. We found that both failure rate and efficiency became poorer as PE prevalence increased, which supported the robustness of our main finding that the performance of each diagnostic strategy became poorer in healthcare settings with higher PE prevalence.

Third, the YEARS algorithm and the Wells rule with PTP-adjusted D-dimer (PeGED) were less safe in this IPD-MA than in their original studies [15,17]. In most of the included studies, the reference standard for PE was a combination of imaging tests and clinical follow-up, with the decision to refer for imaging guided by the diagnostic strategy under evaluation. However, diagnostic strategies adapting D-dimer to PTP, such as YEARS and PeGED, are more efficient than the other strategies. Accordingly, when applying these diagnostic strategies retrospectively in other studies, more patients will have had imaging as the reference standard than clinical follow-up compared to their derivation studies. This approach likely led to the inclusion of small, possibly insignificant clots in the proportion of missed PE cases among those in whom PE could be considered excluded based on a negative PTP-adjusted D-dimer strategy. This hypothesis is supported by data showing that PE detected by the original Wells rule with a fixed D-dimer cutoff included more subsegmental PE than in those detected by the PTP-adjusted YEARS algorithm [30]. Unfortunately, detailed information about the localisation and extent of diagnosed PE was not available in this IPD dataset.

Fourth, as shown in Table D in S1 Text, different types of D-dimer assay were used in the included studies, which could be a source of between-study heterogeneity. In addition, the performance of diagnostic strategies in each healthcare setting could be affected by the variation in D-dimer testing (e.g., the skill of laboratory technicians or the timing of the blood test in relation to patient presentation), which we could not explore in this IPD.

Finally, the studies included in our IPD-MA were conducted between 2000 and 2019. Over those 20 years, the performance of D-dimer testing and imaging studies has evolved. Hence, although we consider the trends of failure rate and efficiency of the diagnostic strategies in our findings to be valid and representative, the validity of our finding in today's patients should be interpreted with some caution.

## Conclusions

The performance of available diagnostic strategies for patients with suspected PE varied considerably across different healthcare settings. The findings of this large-scale study indicate which is the optimum diagnostic strategy for ruling out PE per care setting, balancing the trade-off between missing PE cases and decreasing unnecessary referrals or follow-up.

## Supporting information

**S1 Checklist.** Includes Table A PRISMA-IPD Checklist, Table B PRISMA-DTA Checklist, and Table C PRISMA-DTA for Abstracts Checklist. PRISMA-DTA, Preferred Reporting Items for Systematic Reviews and Meta-Analyses of Diagnostic Test Accuracy; PRISMA-IPD, Preferred Reporting Items for Systematic Reviews and Meta-Analyses of Individual Participant Data.
(DOCX)

**S1 Text.** Includes a detailed statistical analyses plan (including references), Table A Diagnostic strategies under evaluation, Table B Data template, Table C Summary of missing data in each study, and Table D Summary of included studies.
(DOCX)

**S1 Fig.** Includes Fig A Flow of studies, Fig B Risk of bias assessment, Fig C The range of failure rate and efficiency of the diagnostic strategies with $I^2$ statistics, Fig D Sensitivity analysis including only studies in which all diagnostic strategies can be calculated, and Fig E Sensitivity analysis including only studies in which all diagnostic strategies except PERC algorithm can be calculated. PERC, Pulmonary Embolism Rule-out Criteria.
(DOCX)

## Author Contributions

**Conceptualization:** Geert-Jan Geersing, Frederikus A. Klok, Harry R. Büller, Gregoire Le Gal, Jeffrey A. Kline, Menno V. Huisman, Karel G. M. Moons, Marc Righini, Philip S. Wells, Kerstin de Wit, Noémie Kraaijpoel, Nick van Es.

**Data curation:** Geert-Jan Geersing, Toshihiko Takada, Frederikus A. Klok, Harry R. Büller, D. Mark Courtney, Yonathan Freund, Javier Galipienzo, Gregoire Le Gal, Waleed Ghanima, Jeffrey A. Kline, Menno V. Huisman, Karel G. M. Moons, Arnaud Perrier, Sameer Parpia, Helia Robert-Ebadi, Marc Righini, Pierre-Marie Roy, Maarten van Smeden, Milou A. M. Stals, Philip S. Wells, Kerstin de Wit, Noémie Kraaijpoel, Nick van Es.

**Formal analysis:** Geert-Jan Geersing, Toshihiko Takada, Frederikus A. Klok, Karel G. M. Moons, Sameer Parpia, Maarten van Smeden, Milou A. M. Stals, Noémie Kraaijpoel, Nick van Es.

**Funding acquisition:** Geert-Jan Geersing.

**Investigation:** Geert-Jan Geersing, Toshihiko Takada, Frederikus A. Klok, D. Mark Courtney, Yonathan Freund, Javier Galipienzo, Gregoire Le Gal, Waleed Ghanima, Jeffrey A. Kline, Menno V. Huisman, Karel G. M. Moons, Arnaud Perrier, Sameer Parpia, Helia Robert-Ebadi, Marc Righini, Pierre-Marie Roy, Maarten van Smeden, Milou A. M. Stals, Philip S. Wells, Kerstin de Wit, Noémie Kraaijpoel, Nick van Es.

**Methodology:** Geert-Jan Geersing, Toshihiko Takada, Karel G. M. Moons, Sameer Parpia, Maarten van Smeden, Nick van Es.

**Project administration:** Geert-Jan Geersing.

**Resources:** Geert-Jan Geersing.

**Software:** Geert-Jan Geersing, Maarten van Smeden, Nick van Es.

**Supervision:** Geert-Jan Geersing, Frederikus A. Klok, Harry R. Büller, Menno V. Huisman, Karel G. M. Moons, Marc Righini, Maarten van Smeden, Nick van Es.

**Validation:** Geert-Jan Geersing, Maarten van Smeden.

**Visualization:** Geert-Jan Geersing, Maarten van Smeden.

**Writing – original draft:** Geert-Jan Geersing, Toshihiko Takada, Maarten van Smeden, Noémie Kraaijpoel, Nick van Es.

**Writing – review & editing:** Geert-Jan Geersing, Toshihiko Takada, Frederikus A. Klok, Harry R. Büller, D. Mark Courtney, Yonathan Freund, Javier Galipienzo, Gregoire Le Gal, Waleed Ghanima, Jeffrey A. Kline, Menno V. Huisman, Karel G. M. Moons, Arnaud Perrier, Sameer Parpia, Helia Robert-Ebadi, Marc Righini, Pierre-Marie Roy, Maarten van Smeden, Milou A. M. Stals, Philip S. Wells, Kerstin de Wit, Noémie Kraaijpoel, Nick van Es.

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
