## [Editor Report · Decision Letter 0]

3 Sep 2021

Dear Dr Geersing, 

Thank you for submitting your manuscript entitled "Ruling-out Pulmonary Embolism across Different Healthcare Settings: A Systematic Review and Individual Patient Data Meta-Analysis" for consideration by PLOS Medicine.

Your manuscript has now been evaluated by the PLOS Medicine editorial staff and I am writing to let you know that we would like to send your submission out for external peer review.

Please re-submit your manuscript within two working days, i.e. by Sep 07 2021 11:59PM.

Kind regards,

Callam Davidson

Associate Editor

PLOS Medicine

---

## [Decision Letter · Decision Letter 1]

8 Nov 2021

Dear Dr. Geersing,

Thank you very much for submitting your manuscript "Ruling-out Pulmonary Embolism across Different Healthcare Settings: A Systematic Review and Individual Patient Data Meta-Analysis" (PMEDICINE-D-21-03745R1) for consideration at PLOS Medicine. 

Your paper was evaluated by an associate editor and discussed among all the editors here. It was also discussed with an academic editor with relevant expertise, and sent to independent reviewers, including a statistical reviewer. The reviews are appended at the bottom of this email and any accompanying reviewer attachments can be seen via the link below:

[LINK]

In light of these reviews, I am afraid that we will not be able to accept the manuscript for publication in the journal in its current form, but we would like to consider a revised version that addresses the reviewers' and editors' comments. Obviously we cannot make any decision about publication until we have seen the revised manuscript and your response, and we plan to seek re-review by one or more of the reviewers. 

We hope to receive your revised manuscript by Nov 29 2021 11:59PM. Please email us (plosmedicine@plos.org) if you have any questions or concerns.

We look forward to receiving your revised manuscript. 

Sincerely,

Callam Davidson, 

PLOS Medicine

plosmedicine.org

PLOS Medicine requires that the de-identified data underlying the specific results in a published article be made available, without restrictions on access, in a public repository or as Supporting Information at the time of article publication, provided it is legal and ethical to do so. Please see the policy at 

http://journals.plos.org/plosmedicine/s/data-availability

and FAQs at 

http://journals.plos.org/plosmedicine/s/data-availability#loc-faqs-for-data-policy

Please include continuous line numbering throughout the document to facilitate further review.

In the last sentence of the Abstract Methods and Findings section, please describe the main limitation(s) of the study's methodology.

Citations should be in square brackets, and preceding punctuation.

Please provide the appropriate completed PRISMA checklists (IPD and DTA).

When completing the checklists, please use section and paragraph numbers, rather than page numbers.

Please cite the completed checklists in the Methods (e.g. S1 Checklist, or similar). 

Please include additional databases in your search (e.g. Embase, Cochrane Library, Web of Science, ClinicalTrials.gov) or provide justification for the decision to only search MEDLINE.

Please update your search to the present time.

Please provide the name(s) of the institutional review board(s) that provided ethical approval.

Please remove all italics formatting from the References section and only use et al. after listing the first six authors (this applies also to the supplementary references). For further information see https://journals.plos.org/plosmedicine/s/submission-guidelines#loc-references

Comments from the reviewers:

Reviewer #1: This systematic review study examined the performance of different diagnosis approach of PE under different healthcare settings. Instead of pooling estimates, this study used individual level data from each study to estimate the association failure rate and efficiency with different settings for each strategy. Overall, I think the study were well designed and conducted. The methods used to estimate the quantity of interest were solid. 

Below are my specific comments. 

1. Provide line number for easier reference

2. Method, Study eligibility, identification, and selection: why did you not search for other databases such as PubMed for relevant studies? 

3. Method, Study eligibility, identification, and selection: should explain "prospective or cross-sectional design" a bit more. Do you include prospective cohort studies? If the clinical info were only collected at baseline and people develop PE latter, can the clinical information be used to diagnose/predict future PE onset? 

4. Method, Data collection and harmonization: I did not find description of how the data were harmonized. Also, among eligible studies that you sought data from the PI, how many PIs failed to provide data? And is that likely to bias the results (e.g., only PIs of studies with good quality are willing to share data)? 

5. Method, Statistical analyses: I am not sure whether one can examine the between study heterogeneity by calculating the prediction interval. I have not heard of that before. Could you provide citation of such approach. Did you use R "predict" function, predict(…, interval="predict") to obtain the prediction interval? To my understanding, the difference between confidence interval and prediction interval is that the latter further includes randomness of each observation (the random error). I don't think these will help you examine the between study heterogeneity. To examine between study heterogeneity, people usually use intra-cluster correlation (or ICC). 

Reviewer #2: This review addresses and important question concerning a multifactorial diagnostic pathway including imaging and blood testing modalities. The potential variation in the imaging modalities is recognised, but not in the case of the blood test, namely D-dimer. There is known to be significant variation in the analytical and diagnostic performance of the D-dimer test, which is not unexpected when both qualitative and quantitative are available, as well as variation in operator competence - as in the case of point of care testing operators compared with trained laboratory operators. I think points should be recognised in the discussion of limitations as this could have an important impact of the variation between settings. There may be other factors that might impact on performance in different setting, for example in relation to the time at which testing is performed relative to patient presentation.

Reviewer #3: This manuscript presents a 35 000 patient individual patient data meta-analysis seeking to evaluate a number of pulmonary embolism rule out strategies across a wide range of settings. The outcomes are reported in very useful terms as miss rates and efficiencies both with confidence intervals. The review is highly adherent to methodologic and reporting standards across a number of tools endorsed by the EQUATOR network. As an emergency physician I find this paper particularly useful as it provides useful insight as to context-specific performance of the most commonly used clinical prediction rules and approaches to decision-making as it relates to the use of d-Dimer. The limitations section seems transparent and quite complete noting inherent limitations in the data which are well explained and create only minimal threats to validity. Overall, I think this is an ambitious and daunting review that provides useful evaluation of the risk of bias across the 23 included studies and a useful synthesis of the evidence. The conclusions which emphasize the variability in how the performance varies and is to a large degree dependent on the clinical settings is useful and novel guidance that resonates well with me. This work seems to be useful substrate for a clinical guideline effort but this is less well presented in the manuscript.

[LINK]

---

## [Decision Letter · Decision Letter 2]

22 Dec 2021

Dear Dr. Geersing,

Thank you very much for re-submitting your manuscript "Ruling-out pulmonary embolism across different healthcare settings: A systematic review and individual patient data meta-analysis." (PMEDICINE-D-21-03745R2) for review by PLOS Medicine.

I have discussed the paper with my colleagues and the academic editor and it was also seen again by one reviewer. I am pleased to say that provided the remaining editorial and production issues are dealt with we are planning to accept the paper for publication in the journal.

[LINK]

We look forward to receiving the revised manuscript by Jan 05 2022 11:59PM.   

Sincerely,

Callam Davidson, 

Associate Editor 

PLOS Medicine

plosmedicine.org

Requests from Editors:

Lines 76-80: Please update to ‘The main limitation of the primary analyses was that the diagnostic performance of each strategy was compared in different sets of studies since the availability of items used in each diagnostic strategy differed across included studies, however sensitivity analyses suggest the findings were robust’, or similar.

Line 139: ‘The secondary aim’

The titles of Figure S3/S4 appear to be missing some text.

Line 379: ‘Considerably’

Please remove the Strengths and Limitations subheading in your Discussion.

Please remove the Acknowledgements and Data availability sections (the former having no content and the latter being captured in your submission form response).

Comments from Reviewers:

Reviewer #1: I thank the authors for the detailed response to my comments and comments of the editor and other reviewers. I think the revised version looks good. I just have two minor additional comments.

1. Line 232-233: please make sure to include this template in the appendix. Maybe I did not look carefully enough but I did not find this template in the main text or in the appendix.

2. Line 258-260: I see that you are trying to use a new approach to examine the between study heterogeneity. Although I think you justify the method you used, I still believe it would be helpful to also provide the I squared statistics since most people are familiar with that statistics and it's a custom to provide that in a meta-analysis. 

After these two issues are checked/fixed, I think this paper is ready for publication. Congratulations!

[LINK]

---

## [Editor Report · Decision Letter 3]

6 Jan 2022

Dear Dr Geersing, 

On behalf of my colleagues and the Academic Editor, Dr Sanjay Basu, I am pleased to inform you that we have agreed to publish your manuscript "Ruling-out pulmonary embolism across different healthcare settings: A systematic review and individual patient data meta-analysis." (PMEDICINE-D-21-03745R3) in PLOS Medicine.

PRESS

Sincerely, 

Callam Davidson 

Associate Editor 

PLOS Medicine